# Association between Phase Angle from Bioelectric Impedance and Muscular Strength and Power in Physically Active Adults

**DOI:** 10.3390/biology11091255

**Published:** 2022-08-24

**Authors:** Aryanne Hydeko Fukuoka, Núbia Maria de Oliveira, Catarina N. Matias, Filipe J. Teixeira, Cristina P. Monteiro, Maria J. Valamatos, Joana F. Reis, Ezequiel Moreira Gonçalves

**Affiliations:** 1Postgraduate Program in Human Movement Sciences, Center for Health Sciences, State University of North Paraná (UENP), Jacarezinho 86400-000, PR, Brazil; 2TIME, Exercise and Metabolism Investigation Team, UENP, Center for Health Sciences, Jacarezinho 86400-000, PR, Brazil; 3School of Medical Sciences, State University of Campinas (UNICAMP), Campinas 13083-888, SP, Brazil; 4CIDEFES, Universidade Lusófona, 1749-024 Lisboa, Portugal; 5Bettery Life Lab, Bettery S.A., 2740-262 Oeiras, Portugal; 6Interdisciplinary Center for the Study of Human Performance (CIPER), Faculdade de Motricidade Humana, Universidade de Lisboa, Estrada da Costa, 1495-751 Cruz-Quebrada, Portugal; 7Atlântica Instituto Universitário, Department of Nutrition Fábrica da Pólvora de Barcarena, 2730-036 Barcarena, Portugal; 8Laboratory of Physiology and Biochemistry of Exercise, Faculdade de Motricidade Humana, Universidade de Lisboa, 1495-751 Cruz-Quebrada, Portugal; 9Neuromuscular Research Lab, Faculdade de Motricidade Humana, Universidade de Lisboa, Estrada da Costa, 1495-751 Cruz-Quebrada, Portugal; 10Portugal Football School, Portuguese Football Federation, 1495-433 Cruz Quebrada, Portugal

**Keywords:** phase angle, muscle power, muscle strength, body composition

## Abstract

**Simple Summary:**

This study compared muscle strength and power indicators in resistance-trained men by evaluating associations between phase angle and performance, while controlling for body composition. We evaluated dynamic muscle strength in bench press and back squat, and muscle power using the Wingate test and countermovement jump. Participants with higher phase angle displayed superior muscle strength of the upper limbs and greater muscle power of the lower limbs. Regarding countermovement jump and bench press, phase angle showed moderate association with performance, even after controlling for body composition. Still, lean soft tissue was the most important predictor of muscle strength and power.

**Abstract:**

This study aimed to compare muscle strength and power indicators according to bioimpedance spectroscopy’s phase angle (PhA) values, in resistance-trained (RT) men, while exploring associations between PhA and performance. Forty-four men aged 18–45 years, engaged in RT, were allocated according to PhA tertiles. Lean soft tissue (LST) and fat mass (%FM) were assessed using dual-energy x-ray absorptiometry; dynamic muscle strength using 1 repetition maximum (1RM) of bench press (BP) and back squat (BS) and muscle power using Wingate test (WT) and countermovement jump (CMJ). For WT and CMJ, the 3rd tertile was significantly higher than the 1st tertile (*p* = 0.027 and *p* = 0.018, respectively). Regarding BP 1RM, the 3rd tertile was significantly higher than the 2^nd^ tertile (*p* = 0.037). LST better explained the variability in the WT, BS and BP (*p* =< 0.001), while %FM better accounted for jump height in CMJ (*p* =< 0.001). PhA was a predictor of performance in both CMJ (*p* = 0.040) and BP (*p* = 0.012), independently of LST and %FM. Participants with higher PhA also displayed superior muscle strength of the upper limbs and greater muscle power of the lower limbs. PhA displayed significant moderate associations with performance in CMJ and BP, even after controlling for body composition. Still, LST was the most important predictor of muscle strength and power.

## 1. Introduction

The phase angle (PhA) derived from bioelectrical impedance (BI) has been used as an indicator of health [1], body composition [2,3] and physical performance [4,5]. BI is considered a safe method to evaluate PhA, monitor body composition and water compartments. Moreover, it is a portable, non-invasive and inexpensive method, considering each evaluation [6]. Previous studies have shown that PhA is associated with functionality [7], muscle mass quantity [6,8,9] and muscle quality [10], as well as with physical performance in different populations and age groups [11]. More specifically, associations between PhA and knee isokinetic muscle strength and jump performance in adolescent athletes [12], strength of upper and lower limbs in former athletes [6], muscle power of upper limbs [13] and lower limbs in athletes [11], maximum anaerobic power and fatigue index [14] and sprint speed [15] in soccer athletes, were observed. In brief, increases in PhA have been related to increases in performance.

Resistance training (RT) is safe and highly recommended for the general population of all ages, when aiming to enhance strength and muscle mass, to improve body composition and physical fitness and to prevent several chronic diseases [16,17]. Research in older adults has shown that RT increases PhA [10,18,19,20,21,22,23,24,25,26,27]; with similar results being observed in obese women [28], RT-trained women [29] and untrained young adults of both genders [30]. In addition, another study [31] conducted with young males (18.8 ± 0.5 years) showed improvements in PhA after 6 months of military training.

Moreover, PhA has been associated with performance variables in the athletic population [11,12,13,14,15] and in the elderly [20,25,26]. However, to the best of our knowledge, this association has not yet been observed or even tested in active adults engaging in RT. Additionally, these studies evaluated mainly isometric muscle strength through handgrip strength tests [11,13,18,28]. It is also important to state that more than isometric strength, dynamic strength and muscle power are widely used daily in regular activities and sports actions, and exercises such as back squats and bench press are widely used in RT programs. Thus, the analysis of a possible association between PhA (assessed through a simple, safe, fast and non-invasive method such as BI) and performance regarding these exercises can add to the current body of literature insofar as training monitoring is concerned. This study aimed to (a) compare the muscle strength and power assessments according to PhA values (tertiles) in RT-experienced men and (b) explore the associations between PhA and performance variables controlling for body composition variables.

## 2. Materials and Methods

### 2.1. Participants and Study Design

This cross-sectional study was a part of a double-blind randomized controlled trial; details of the study can be found elsewhere [32].

This investigation was approved by the Faculty of Human Kinetics Institutional Review Board (approval number 15/2017) and conformed to all standards of human research set out in the Declaration of Helsinki. Participants were informed about the objectives and procedures of the study and signed the informed consent form. The trial was registered at ClincicalTrials.gov as NCT03511092.

All apparently healthy individuals between 18 and 45 years, currently engaged in resistance training for at least one year and at least three times per week were considered eligible for the study. Exclusion criteria included individuals taking any type of drug (including anabolic steroids), medicines or supplements that may enhance body composition or performance, 3 months prior to the research. Smoking and having a diagnosed disease that could compromise the tolerance to the supplements used in the original study or influence body composition also lead to exclusion. The inclusion and exclusion criteria were previously reported [32]. Participants were recruited from social networks and local gyms. Initially, 53 men were recruited according to the eligibility criteria. Nine individuals were excluded since they failed to present all the results of the bioimpedance parameters and/or performance tests. The final set of this study consisted of 44 volunteers. The sample size for the original research study [33] was calculated through an a priori power analysis (G*Power Version 3.1.9.2, Heinrich Heine Universitat Dusseldorf, Dusseldorf, Germany), based on FFM changes from previous studies and power of 0.80 and alpha of 0.05. A total of 44 participants were recommended and an additional 20% (9 participants) were recruited to account for possible drop out.

The participants performed a familiarization session within one week of the evaluation, comprising the muscle strength and power assessments.

On the evaluation day, participants reported to the laboratory after a 12 h fast, without prior exercise and having refrained from the consumption of alcohol in this period. Participants were instructed to not perform exercise in the 12 h before the testing session. Body composition by dual-energy X-ray absorptiometry (DXA) and bioimpedance spectroscopy analysis (BIS) was performed in this order, in the morning (7–9 AM). Afterwards, after the ingestion of a meal replacement bar and a 5 min warm up of unloaded cycling on a cycle ergometer (60–70 rpm); countermovement jump (CMJ), dynamic muscle strength and Wingate evaluations were performed in this order, with a minimum of 10 min passive rest in between.

### 2.2. Body Composition and Bioimpedance Measurements 

Height and weight were recorded to the nearest 0.1 cm and 0.1 kg (Seca model 877, Hamburg, Germany), respectively. Body mass index (BMI) was determined by the ratio between weight (kg) and squared height (m^2^). Whole-body bone mineral content (BMC, g), fat mass (FM, %) and lean soft tissue (LST, kg) were obtained by DXA on a Hologic Explorer-W fan-beam densitometer (Hologic, Waltham, MA, USA) according to the standard procedures described previously [32]. The same technician positioned the participants, performed the scans and executed the analysis according to the operator’s manual using the standard analysis protocol.

Before the BIS evaluation, each subject rested in a supine position for ten minutes to stabilize body fluids. Afterwards, the measurements were made with the subjects in the supine position with a leg opening of approximately 45 degrees compared to the median line of the body and the upper limbs positioned about 30 degrees away from the trunk. After the skin was cleaned with alcohol, four electrodes were placed on the dorsal surfaces of the right hand and foot. The source electrodes were placed on the hand, in the middle of the dorsal surface proximal to the metacarpal-phalangeal joint, and on the foot, in the middle of the dorsal surface proximal to the metatarsal-phalangeal joint. The detector electrodes were placed on the wrist at the midline between the distal prominences of the radius and ulna, and in the ankle joint at the line between the malleoli. The participants were asked to empty their bowels and bladder at most 30 min before the measurement. BIS (model 4200 Xitron Technologies, San Diego, CA, USA) was used to assess extracellular (ECW) and intracellular (ICW) water, as previously stated by our group [33], and the ECW/ICW ratio was determined accordingly [34]. From the raw data, resistance (R) and reactance (Xc) at a frequency of 50 kHz, PhA was calculated as the arctangent of Xc/R × 180°/π [35]. Fan beam equipments, when compared to four compartment models as criterion, present low bias (less the 1% for body fat), although at an individual level, higher differences from the reference method have been reported [36]. Regarding our specific DXA model, a validation study was performed, and reported the same bias as the general fan beam densitometers [37]. In our laboratory, in ten healthy adults, the test–retest coefficient of variation for both FM and FFM is 0.8% and 1.7%, respectively, and for TBW, ECW and ICW, is 0.3%, 0.7% and 0.3%, respectively.

### 2.3. Dynamic Muscle Strength

Dynamic muscle strength was assessed by one repetition maximum (1RM) of the bench press and back squat exercises. The evaluation of 1RM was carried out on a multipower machine (Model-M953; Technogym, Cesena, Italy) according to the National Strength and Conditioning Association (NSCA) guidelines and was supervised by an NSCA-certified strength and conditioning specialist. In summary, the subjects performed 3 warmup sets before attempting the 1RM load. A 3-min recovery was allowed between the last warmup set and the 1RM attempt. An increase or decrease of 2.5–5 kg on the bench press exercise and 5–10 kg on the squat exercise occurred in case the attempt to move the 1RM load was successful or failed. The order of the 1RM exercise determination was: bench press and back squat.

### 2.4. Muscle Power

Muscle power was assessed by a supramaximal cycling test (Wingate) using a cycle ergometer (Monark ergomedic 894 E, Monark Exercise AB; Vansbro, Sweden), and CMJ, using a contact platform controlled by an open-source hardware and software model (Chronojump Boscosystems; Barcelona, Spain), as previously described [31]. 

After individual adjustment on the cycle ergometer, participants performed three to five minutes of light cycling. At the end of each minute of the warm-up, the participant performed approximately five seconds of sprinting. During the Wingate test, the volunteers were instructed to cycle against a predetermined resistance (7.5% body weight) as fast as possible for 30 s. Wingate Average Power (W) was calculated (average of all five-second intervals over the entire 30 s test). 

Countermovement jump height was assessed, computing flight time with a temporal resolution of 1 ms. After a minimum of two practice attempts until the results were consistent, participants completed three CMJ for evaluation with two minutes rest in between. The best attempt (cm) out of the three was considered for analysis. Participants were instructed to keep the arms akimbo and jump as high as possible while making sure to land back on the contact mat. Participants performed the jumping tests using a self-selected depth of crouch (which approximated to a half squat) in their own time. Legs were required to remain fully extended throughout the flight phase of the CMJ. Visual inspection was used to ensure hands remained in contact with the hips during each trial and that the subject posture was the same at take-off and landing.

## 3. Statistical Analysis

Data analysis was performed using the IBM SPSS Statistics (SPSS Inc., IBM Company, Chicago, IL, USA) version 25.0. Descriptive statistics including mean, standard deviation, standard error of mean, median, minimum and maximum values were performed for all outcome measurements. All variables were tested for normality using the Shapiro–Wilk test. Variables that did not meet the normality criteria were transformed by log10 (BMI, TBW, ICW, %FM), BMC, Wingate’s average power and bench press 1RM. To verify the influence of PhA on muscle strength and power, participants were divided into three groups according to PhA tertiles (1st tertile ≤ 8.18°, *n* = 15; 2nd tertile between 8.18° and 8.90°, *n* = 15; and 3rd >8.90°, *n* = 14). Comparison of the sample characteristics according to PhA tertiles was analysed using one-way ANOVA. Comparisons of results of performance tests between tertiles groups of PhA were performed by Covariance analysis (ANCOVA) controlling for age and %FM. The post-hoc analysis was conducted using Bonferroni’s correction. The stepwise linear regression method was used to determine the best predictor for muscular strength and power performance. Linear regression analysis was used to check the potential of PhA as an independent predictor in the performance tests, and after adjusting for LST (Model 1), and LST and %FM (Model 2) as covariables. The significance level for α was set at *p* ≤ 0.05.

## 4. Results

The participants from the lowest tertile of PhA (1st tertile) displayed significantly higher values for age (*p* = 0.037) and %FM (*p* = 0.025) and significantly lower values for ICW (*p* = 0.003) when comparing to the participants from the highest tertile (3rd tertile) (Table 1). The participants from the 1st tertile also showed significantly higher extra/intracellular water ratio values than participants from the 2nd and 3rd tertiles (*p* < 0.001 for both).

Regarding muscle power performance (adjusted for age and %FM), participants in the 3rd tertile displayed significantly higher values of average power, determined using the Wingate test (*p* = 0.038), and jump height (cm) from the CMJ (*p* = 0.022), compared with the 1st tertile. For muscle strength, a significant difference in bench press 1RM, between the 3rd and 2nd tertiles (*p* = 0.038) was observed. No significant difference was observed between tertiles, regarding lower limb muscle strength (back squat 1RM) (Table 2).

A stepwise regression was performed to identify the predictor variables most associated with performance in each test. Figure 1 illustrates that LST was the variable that best explained the variability of the results in the Wingate test, back squat and bench press 1RM tests, while %FM determined CMJ performance.

Regarding performance variables, PhA was a significant independent predictor of CMJ (β = 0.49; *p* = 0.001) and bench press 1RM (β = 0.55; *p* < 0.001), even after controlling for LST and %FM (CMJ: Model 1: β = 0.42; *p* = 0.005 and Model 2: β = 0.30; *p* = 0.040; bench press 1RM: Model 1: β = 0.36; *p* = 0.002 and Model 2: β = 0.29; *p* = 0.012). In the Wingate test, the explanatory power of PhA (β = 0.36; *p* = 0.017) was shown to be dependent on body composition, losing significance in both adjusted models 1 and 2 (*p* = 0.295 and *p* = 0.169, respectively). Regarding the dynamic strength of lower limbs (back squat 1RM), PhA showed no association (Table 3).

## 5. Discussion

In this sample of men with RT experience, we observed that the group with the highest PhA values (3rd tertile) had higher values of muscle power (Wingate and CMJ tests) and dynamic strength of upper limbs (bench press 1RM). Although the variables that best predicted the variability of the results in the performance tests were LST (Wingate test, bench press 1RM and back squat 1RM) and %FM (CMJ), PhA was also directly associated with the power output from the CMJ and the dynamic strength resulting from the bench press, independently of LST and body fat. 

The direct association between PhA and muscle power observed in the present study has been previously reported. Namely, in soccer players, changes in regional PhA were strongly associated with changes in CMJ performance [38] and other types of cross-sectional tests, such as 10 and 30 m sprints [15]; Nabuco and colleagues [14] used the Running Anaerobic Sprint Test (RAST), which showed a direct association between PhA with mean power across sprints, even after being adjusted for fat free mass and body fat. From these previous works, one could suggest that PhA may be a valid inverse predictor of fatigue. 

For muscle strength, we observed direct associations between PhA and bench press 1 RM. These results are in line with a previous investigation that observed a direct association of PhA with muscle performance, assessed via handgrip strength test, in adult athletes of several modalities [11]. Futhermore, isometric strength has been shown to have a strong association with PhA in both athletes [13] and former athletes [6]. In healthy elderly women, the dynamic strength of the lower limbs was assessed after six months of RT using five repetitions (5RM) of the leg extension strength test, with a direct association being reported between PhA and leg extension 5RM [20].

These results may be related to the fact that greater PhA reflects greater cellular integrity, ICW, muscle mass [39] and muscle quality [10,25,26], probably related with muscle composition and fiber structure, explaining the direct associations of PhA with performance [15,39], and reflecting the effects of life long sports practice [6,40]. 

In support of this hypothesis, in our study, not only did the 3rd PhA tertile present higher values for ICW than the 1st tertile, but also, PhA tertiles were able to differentiate the ECW/ICW ratios, in which the PhA values were inversely proportional to the ECW/ICW ratio (3rd tertile > 2nd tertile > 1st tertile). Similar results were observed in athletes, regardless of other factors, such as body composition, age, height and sport category [39]. The ICW is located mainly in muscle mass and ECW can be an indicator of edema; thus, PhA has been previously reported as a valid inverse indicator of the ECW/ICW ratio [18].

The observed associations between PhA and performance variables are in accordance with the significant differences observed between PhA tertiles for muscle power (Wingate and CMJ tests) and dynamic strength of the upper limbs (bench press 1RM) when adjusted for age and FM, where the 3rd tertile presented higher values for muscle power variables than the 1st, and higher values for dynamic strength of upper limbs than the 2nd tertile. The fact that the 3rd tertile was not higher than the 1st for bench press 1RM may be related to training factors such as frequency and objective, among others.

Although PhA is able to significantly explain some variability regarding the studied performance variables, in this cross-sectional study, LST was pointed out as the variable that best explained muscle power (Wingate test) and dynamic strength (bench press 1RM and squat 1RM), emphasizing its importance as a predicting variable of performance. These results agree with others that have identified the amount of muscle mass as a predictor of strength and power performance in young athletes [41]; and have observed a direct association of LST with sports performance [15] and cardiorespiratory fitness [42].

Despite these interesting results, our study has some limitations that should be considered. First, our sample size was small; the second limitation is the study design, which is cross-sectional and cannot establish a cause–effect relationship. However, our study also some strengths that should be addressed, namely that this study is unprecedented, in which our sample was comprised of trained participants (with at least 1 year of resistance training) with a homogeneous sample, including PhA values (all sample: minimum and maximum value: 7.32–10.34) that, even after allocation into groups, was still able to predict the level of performance in each tertile. Additionally, the assessment of body composition was not determined from predictive equations using bioimpedance parameters, which could increase the error and influence the association analysis pertaining to PhA. Instead, we used DXA, a method that shows high validity and reproducibility regarding the assessment of body composition.

## 6. Conclusions

In this sample of RT trained men, participants with higher PhA values displayed better performance regarding lower limb muscle power and upper limb strength tests. PhA showed a direct association with CMJ power and bench press 1RM strength, even after controlling for body composition. Additionally, body composition, especially LST, was the most important predictor of muscle strength and power in this sample.

## Figures and Tables

**Figure 1 biology-11-01255-f001:**
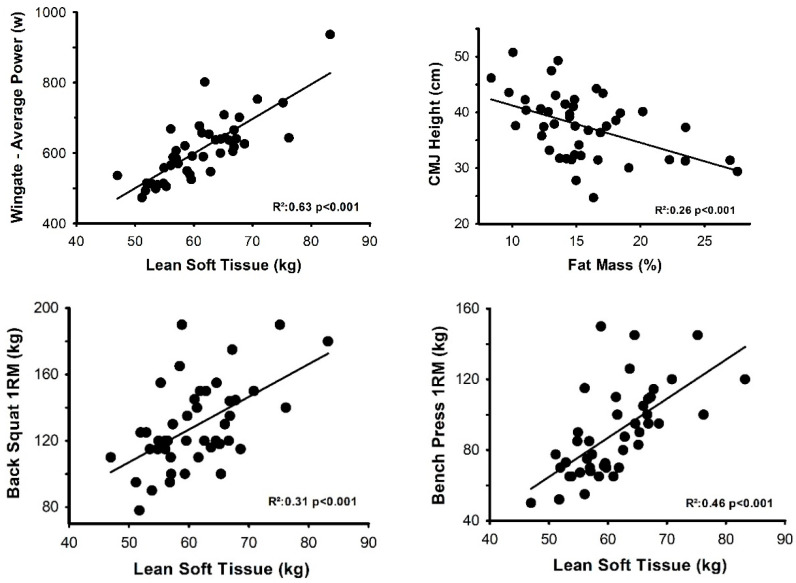
Scatter plots of the associations between performance results in the tests (Wingate test: **upper left** panel, CMJ “counter movement jump” height: **upper right** panel, Back squat: **lower left** panel, Bench press: **lower right** panel).

**Table 1 biology-11-01255-t001:** General characteristics of the sample, body composition and BIS parameters according to PhA tertiles.

		1st Tertile n = 15	2nd Tertile n = 15	3rd Tertile n = 14	F	*p*-Value
Age (years)	Mean	34.3	30.8	27.4 *	3.43	0.042
	SD	6.2	8.2	6.6		
Weight (kg)	Mean	77.0	73.9	79.9	1.60	0.214
	SD	9.8	5.8	10.8		
Height (cm)	Mean	173.8	173.3	174.7	0.23	0.792
	SD	6.5	4.4	6.1		
BMI (kg/m^2^)	Median	25.4	24.6	25.7	2.72	0.079
	Min–Max	22.0–29.3	21.5–26.6	22.6–33.3		
LST (kg)	Mean	59.3	59.8	64.5	2.36	0.107
	SD	7.2	5.4	8.2		
FM %	Median	16.3	14.5	13.2 *	5.18	0.010
	Min–Max	12.3–27.6	8.4–18.1	9.7–20.2		
BMC (kg)	Median	2.7	2.8	2.9	2.33	0.110
	Min–Max	2.2–4.0	2.4–3.4	2.5–4.0		
ECW (kg)	Mean	19.6	19.3	19.7	0.093	0.911
	SD	2.0	1.9	3.4		
ICW (kg)	Median	29.1	32.5	35.2 *	6.32	0.004
	Min–Max	23.7–36.9	24.8–37.0	24.5–51.1		
ECW/ICW	Mean	0.65	0.60 *	0.54 *^#^	59.33	<0.001
	SD	0.03	0.02	0.03		

BMI: body mass index; LST: lean soft tissue; FM: fat mass; BMC: bone mineral content; ECW: extracellular water; ICW: intracellular water; ECW/ICW: extracellular/intracellular water ratio. * Significantly (*p* < 0.05) different from 1st tertile. ^#^ Significantly (*p* < 0.05) different from 2ndtertile.

**Table 2 biology-11-01255-t002:** Comparison of performance in muscle power and strength tests between PhA tertiles.

	Mean (SEM)			F	*p*-Value
Variable	1st Tertile	2nd Tertile	3rd Tertile		
Muscle Power					
Wingate Average Power (W)	574.5 (24.3)	591.3 (22.6)	667.0 (24.0) *	3.982	0.027
Countermovement jump height (cm)	35.2 (1.4)	36.8 (1.3)	40.9 (1.4) *	4.486	0.018
Muscle Strength					
Back Squat 1RM (kg)	127.8 (7.0)	119.9 (6.6)	140.0 (7.0)	2.297	0.114
Bench Press 1RM (kg)	84.5 (6.2)	80.7 (5.8)	104.0 (6.1) ^#^	3.579	0.037

1RM: one repetition maximum; Values, adjusted for age and %FM, are presented as mean (SEM), SEM being the standard error of the mean. * Significantly (*p* < 0.05) different from the 1st tertile. ^#^ Significantly (*p* < 0.05) different from the 2nd tertile.

**Table 3 biology-11-01255-t003:** Associations between phase angle and performance in power and muscle strength tests.

**Wingate Test ^a^**						
	B	S.E	β	*p*-value	r² ajust.	*p*-value
Phase Angle	0.03	0.01	0.36	0.017	0.11	0.017
Model 1	0.01	0.01	0.10	0.295	0.63	<0.001
Model 2	0.01	0.01	0.15	0.169	0.63	<0.001
**Countermovement Jump**						
	B	S.E	β	*p*-value	r² ajust.	*p*-value
Phase Angle	3.96	1.09	0.49	0.001	0.22	<0.001
Model 1	3.42	1.15	0.42	0.005	0.24	0.001
Model 2	2.44	1.15	0.30	0.040	0.32	<0.001
**Bench Press 1RM ^a^**						
	B	S.E	β	*p*-value	r² ajust.	*p*-value
Phase Angle	0.09	0.02	0.55	<0.001	0.28	<0.001
Model 1	0.06	0.02	0.36	0.002	0.55	<0.001
Model 2	0.05	0.02	0.29	0.012	0.58	<0.001
**Back Squat 1RM**						
	B	S.E	β	*p*-value	r² ajust.	*p*-value
Phase Angle	8.94	5.37	0.25	0.103	0.04	0.103
Model 1	2.63	4.94	0.07	0.597	0.28	<0.001
Model 2	0.54	5.23	0.01	0.919	0.28	0.001

Model 1 = adjusted for LST; Model 2 = adjusted for LST and %FM. ^a^ Log-transformed dependent variable values.

## Data Availability

Not applicable.

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
