# Peer review of "Association between Phase Angle from Bioelectric Impedance and Muscular Strength and Power in Physically Active Adults"

_biology, 2022, doi:10.3390/biology11091255_

Round 1

Reviewer 1 Report

The paper is written well and clear for the readers. Well also the structure (introducution, material methods, results and conclusion). The scientific soundness is good and interesting. best wishes 

Author Response

Dear reviewer,

We would like to thank the reviewer for his/her appreciations.

Best wishes.

Reviewer 2 Report

Manuscript ID: biology-1867731

Title: Association between Phase Angle from Bioe-Lectric Impedance and Muscular Strength and Power in Physically Active Adults

Authors: Aryanne Hydeko Fukuoka, Nubia Maria Oliveira, Catarina N. Matias, Filipe J. Teixeira, Cristina P. Monteiro, Maria J. Valamatos, Joana F. Reis, Ezequiel Moreira Gonçalves

The manuscript by Aryanne Hydeko Fukuoka, Nubia Maria Oliveira, Catarina N. Matias, Filipe J. Teixeira, Cristina P. Monteiro, Maria J. Valamatos, Joana F. Reis and Ezequiel Moreira Gonçalves „Association between Phase Angle from Bioe-Lectric Impedance and Muscular Strength and Power in Physically Active Adults” describe research for the purpose of comparison muscle strength and power indicators according to bioimpedance spectroscopy phase angle values, in 44 resistance-trained men.

The researches carried out are very interesting. The results are good described, but I suggest extending the information about the patients involved in the experiment.

Did the study take into account the diet, drugs or addictions that could significantly affect the result of the study?

What type of randomization was used?

In the introduction, the author wrote that „In Resistance training (RT) is safe and highly recommended for the general population of all ages, when aiming to enhance strength and muscle mass, improve body composition and physical fitness and to prevent several chronic diseases [16, 17]. Research in older adults has shown that RT increases PhA [10,18-27]; with similar results being observed in obese women [28], RT-trained women [29], and untrained young adults of both genders [30]. Moreover, Pha has been associated with performance variables in the athletic population [11-15] and in the elderly [20,25,26]”. This means that previously studies were only conducted on women (all ages), young (<18 years old) and old (> 60 years old) people of both sexes? What about men? No studies have been conducted so far in the age range of 19-59?

Author Response

REVIEWER 2

The manuscript by Aryanne Hydeko Fukuoka, Nubia Maria Oliveira, Catarina N. Matias, Filipe J. Teixeira, Cristina P. Monteiro, Maria J. Valamatos, Joana F. Reis and Ezequiel Moreira Gonçalves “Association between Phase Angle from Bioe-Lectric Impedance and Muscular Strength and Power in Physically Active Adults” describe research for the purpose of comparison muscle strength and power indicators according to bioimpedance spectroscopy phase angle values, in 44 resistance-trained men.

The researches carried out are very interesting. The results are good described, but I suggest extending the information about the patients involved in the experiment.

Answer: We thank the reviewer for the time and effort to help us improve the quality of our manuscript. Below are point-by-point responses to your comments. We addressed all issues in blue throughout the manuscript document.

Did the study take into account the diet, drugs or addictions that could significantly affect the result of the study?

Answer: As stated in our paper (1)” Participants taking any type of medication or supplements aimed at enhancing body composition or performance before the research, were excluded (only protein supplements and multivitamins were allowed).” This Information was included in the manuscript.

What type of randomization was used?

Answer: Thank you for your comment. In this specific study, no type of randomization was used. The purpose of the study was to compare the indicators of muscle strength and power in resistance trained men with different phase angle values, so we chose to divide the groups by tertiles of the phase angle derived from bioelectrical impedance. Nevertheless, if the reviewer was talking about the original research, in which the aim was to evaluate the effects of off-the-shelf leucine metabolite supplements of alpha-HICA, HMB-FA, and HMB-Ca on resistance training-induced changes in muscle thickness and performance, the assignment was according to a randomly generated list and was blocked in varying block sizes with participants matched for grip strength, age, and dual-energy x-ray absorptiometry (DXA)- measured fat-free mass (FFM). Thus, at baseline, there were no statistically significant differences between groups for handgrip strength, age, or FFM (1).

In the introduction, the author wrote that “In Resistance training (RT) is safe and highly recommended for the general population of all ages, when aiming to enhance strength and muscle mass, improve body composition and physical fitness and to prevent several chronic diseases [16, 17]. Research in older adults has shown that RT increases PhA [10,18-27]; with similar results being observed in obese women [28], RT-trained women [29], and untrained young adults of both genders [30]. Moreover, Pha has been associated with performance variables in the athletic population [11-15] and in the elderly [20,25,26]”. This means that previously studies were only conducted on women (all ages), young (<18 years old) and old (> 60 years old) people of both sexes? What about men? No studies have been conducted so far in the age range of 19-59?

Answer: Thank you for your comment. The information about resistance training and PhA including men that we found in the literature had already been cited in the text with the study conducted by Ribeiro et al. (2017). In fact, there is a scarcity of studies conducted with men analyzing the PhA response to resistance training or its association with performance variables. There is a study conducted with a sample of army cadets (18.8 ± 0.5 years) (2) which investigated the PhA response after 6 months of military training, we have included this information in the introduction section.

  1. Teixeira, F. J.; Matias, C. N.; Monteiro, C. P.; Valamatos, M. J.; Reis, J. F.; Tavares, F.; Batista, A.; Domingos, C.; Alves, F.; Sardinha, L. B.; Phillips, S. M. Leucine Metabolites Do Not Enhance Training-induced Performance or Muscle Thickness. Med Sci Sports Exerc 2019, 51, 56–64. DOI: 10.1249/MSS.0000000000001754.
  2. Langer RD, Silva AM, Borges JH, Cirolini VX, Páscoa MA, Guerra-Júnior G, et al. Physical training over 6 months is associated with improved changes in phase angle, body composition, and blood glucose in healthy young males. Am J Hum Biol. 2019;31(5):1–8.

Reviewer 3 Report

Dear Authors

You have written an interesting research. However, some parts need to be addressed for greater clarity and reproducibility.

A simple summary needs to be shortened where you highlight just the main findings in a more applicable way. Please rewrite

Abstract-report p values

The introduction is focused and on point with relevant literature and clearly leads to the main study rationale.

Methods:

Line 96 - correct thrice to three times per week

How was your sample size determined (G*Power or any other method, please report)?

Please briefly describe inclusion and exclusion criteria and not just a reference to a connected study.

Line 103 - correct to Seca and report the exact model of equipment used for measuring height and weight.

Report at what time of the day were body composition measurements done. What was the order of them? What was the break in between? What were the instructions to the participants 1 day before the measurements as this can have a significant impact on measurements? What were the instructions before the measurements? Please be specific. I would suggest looking at the description of the following paper for the BIA measurements (2.3. Experimental Procedure): https://www.mdpi.com/2079-7737/10/11/1199    Please try to describe it in these form.

Please report reliability and validity data for the DXA and bioimpedance used.

Add a reference to the PhA formula.

What was the order of the performance tests? What was the break between them? What was the warm-up? Was it supervised or left to the participants? Please add this info

Was there any demonstration before the tests?

Report the exact model of the equipment used for CMJ measurement (which contact platform???). What were the instructions to participants? Any trial attempts. What was the break in between, and what variables were taken into further analysis? Add info

What variables were taken into further analysis in the Wingate test? Add info

Line 142-143 / PhA - tertiles - please add a reference for this decision and elaborate on why to divide them into tertiles.

In general, the paper is well written and the discussion is on point, however, the methods part still needs some more work from the authors.

Therefore I recommend a major revision.

Kind regards

Author Response

REVIEWER 3

Dear Authors

You have written an interesting research. However, some parts need to be addressed for greater clarity and reproducibility.

Answer: We appreciate the reviewers’ feedback; your input has helped to improve the quality and readability of the manuscript. Below are point-by-point responses to your comments. We addressed all issues in blue throughout the manuscript document.

A simple summary needs to be shortened where you highlight just the main findings in a more applicable way. Please rewrite

Answer: The simple summary was shortened. 

Abstract-report p values

Answer: The p-values were included in the abstract.

The introduction is focused and on point with relevant literature and clearly leads to the main study rationale.

Answer: Thank you for your comment.

Methods:

Line 96 - correct thrice to three times per week

Answer: Corrected.

How was your sample size determined (G*Power or any other method, please report)?

Answer: Sample size was calculated through an a priori power analysis (G*Power Version 3.1.9.2, Heinrich Heine Universitat Dusseldorf, Germany), based on FFM changes from previous studies and power of 0.80 and alpha of 0.05, as stated in the original research study (1). This Information was included in the manuscript.

Please briefly describe inclusion and exclusion criteria and not just a reference to a connected study.

Answer: Information was included in the manuscript as requested.

Line 103 - correct to Seca and report the exact model of equipment used for measuring height and weight.

Answer: The word was corrected and the model was reported.

Report at what time of the day were body composition measurements done. What was the order of them? What was the break in between? What were the instructions to the participants 1 day before the measurements as this can have a significant impact on measurements? What were the instructions before the measurements? Please be specific. I would suggest looking at the description of the following paper for the BIA measurements (2.3. Experimental Procedure): https://www.mdpi.com/2079-7737/10/11/1199    Please try to describe it in these form.

Answer: We acknowledge that the methods needs more detailed description.  More detailed information on the methods were added to the manuscript in “Participants and study design” and “Body composition and bioimpedance measurements” that we think answer the reviewer’s concerns.

Please report reliability and validity data for the DXA and bioimpedance used.

Answer: Information was included in the manuscript.

Add a reference to the PhA formula.

Answer: The reference was added.

What was the order of the performance tests? What was the break between them? What was the warm-up? Was it supervised or left to the participants? Please add this info

Was there any demonstration before the tests?

Report the exact model of the equipment used for CMJ measurement (which contact platform???). What were the instructions to participants? Any trial attempts. What was the break in between, and what variables were taken into further analysis? Add info

Answer: Information was included in the manuscript. Please see “Participants and study design”; “Dynamic Muscle strength” and “Muscle Power” sections.

What variables were taken into further analysis in the Wingate test? Add info

Answer: Only Wingate Average Power (W), as stated in table 2. Information was included in the methods section.

Line 142-143 / PhA - tertiles - please add a reference for this decision and elaborate on why to divide them into tertiles.

Answer: In order to investigate whether the phase angle values ​​could discriminate different levels of performance in the tests, we chose to separate the sample by PhA tertiles.

In general, the paper is well written and the discussion is on point, however, the methods part still needs some more work from the authors.

We appreciate your revision that improved the overall quality of the manuscript.

Therefore I recommend a major revision.

Kind regards

Best regards

References

  1. Teixeira, F. J.; Matias, C. N.; Monteiro, C. P.; Valamatos, M. J.; Reis, J. F.; Tavares, F.; Batista, A.; Domingos, C.; Alves, F.; Sardinha, L. B.; Phillips, S. M. Leucine Metabolites Do Not Enhance Training-induced Performance or Muscle Thickness. Med Sci Sports Exerc 2019, 51, 56–64. DOI: 10.1249/MSS.0000000000001754.

Round 2

Reviewer 3 Report

Dear Authors

Thank you for addressing the majority of my comments. The paper quality improved.

However, there are some samll parts that need further amendment.

Sample size determination - nice that you included the G*Power sentence but, the main point is missing - what was the calculated-recommended sample size? Please report it.

In the body composition measurement section, there is nothing about what instructions were given 1 day prior to testing in regards to physical activity as this can have a significant effect on body composition, especially on BIS measurements. If none were you need to add this in the limitations section. 

You reported just reliability data from your lab on 10 participants which is not what I asked for - validity is missing. Reliability and validity studies are done on a bigger sample. So please report this from reliability and validity studies done for your equipment.

Kind regards

Author Response

We thank the reviewer for the time and effort to help us improve the quality of our manuscript. Below are point-by-point responses to your comments. We addressed all issues in blue throughout the manuscript document.

Reviewer:

Thank you for addressing the majority of my comments. The paper quality improved.

However, there are some small parts that need further amendment.

Sample size determination - nice that you included the G*Power sentence but, the main point is missing - what was the calculated-recommended sample size? Please report it.

Answer: We added the information in the manuscript.

In the body composition measurement section, there is nothing about what instructions were given 1 day prior to testing in regards to physical activity as this can have a significant effect on body composition, especially on BIS measurements. If none were you need to add this in the limitations section.

Answer: Thank you for the observation. We added a sentence that highlighted that the participants were instruct to not exercise in the 12 h prior to the testing session.

You reported just reliability data from your lab on 10 participants which is not what I asked for - validity is missing. Reliability and validity studies are done on a bigger sample. So please report this from reliability and validity studies done for your equipment.

Answer: We added the required information in the manuscript adding references pertaining the validity of your equipment. Please see: “Body composition and Bioimpedance measurements” section.